# Underweight predicts extubation failure after planned extubation in intensive care units

**Chung-Yeh Chuang[1,2], Han-Shui Hsu** [1,3⊗] *, **Guan-Jhou Chen** [4,5⊗] *, **Tzu-Yi Chuang[6], Ming-Han Tsai[2]**

**1** Institute of Emergency and Critical Care Medicine, National Yang-Ming Chiao Tung University, Taipei, Taiwan, **2** Department of Critical Care Medicine, Min-Sheng General Hospital, Taoyuan, Taiwan, **3** Department of Thoracic Surgery, Taipei Veterans General Hospital, Taipei, Taiwan, **4** Department of Internal Medicine, National Taiwan University College of Medicine, Taipei, Taiwan, **5** Infection Control Room, Min-Sheng General Hospital, Taoyuan, Taiwan, **6** Department of Chest Medicine, Min-Sheng General Hospital, Taoyuan, Taiwan

⊗ These authors contributed equally to this work.
* hsuhs@vghtpe.gov.tw(H-SH); guanjhouchen@ntu.edu.tw (G-JC)

**Data Availability Statement:** All relevant data are within the paper and its Supporting Information files.

**Funding:** The study did not receive any funding for public or private sectors.

## Abstract

### Background

Body weight is associated with different physiological changes and the association between weight and mortality in critical care setting had been discussed before. In this study, we investigated the linkage between underweight and post-extubation failure in mechanical ventilated patients in critical setting.

### Methods

This is a retrospective cohort study including patients who were admitted to medical or surgical intensive care units (ICU) between June 2016 and July 2018 and had received endotracheal intubation for more than 72 hours. Those who passed spontaneous breathing trial and underwent a planned extubation were enrolled. Extubation failure was defined as those who required reintubation within the first 72 hours for any reasons. The probability of extubation failure was calculated. Demographic and clinical characteristics were recorded. Multivariate logistic regression models were then used to determine the potential risk factors associated with extubation failure.

### Results

Overall, 268 patients met the inclusion criteria and were enrolled in our study for analysis. The median age of included patients was 67 years (interquartile range, 55–80 years) with 65.3% being male; 63.1% of the patients were included from medical ICU. The proportion of extubation failure in our cohort was 7.1% (19/268; 95% confidence interval [CI], 4.3–10.9%). Overall, underweight patients had the highest risk of extubation failure (8/50), as compared with normoweight (9/135) and overweight patients (2/83). In the multivariate analysis, being underweight (adjust OR [aOR], 3.80, compared to normoweight; 95% CI, 1.23–11.7) and

**Competing interests:** The authors have declared that no competing interests exist.

lower maximal inspiratory airway pressure (aOR per one $cmH_2O$ decrease, 1.05; 95% CI 1.00–1.09) remained significantly associated with extubation failure.

## Conclusion

In our study, being underweight and lower maximal inspiratory airway pressure was associated with post-extubation respiratory failure after a planned extubation.

## Introduction

Weaning from mechanical ventilation and extubation are important decisions in intensive care units (ICUs). Inappropriate timing of extubation and weaning off the ventilator might lead to unnecessary invasive procedures for reintubation and deleterious complications. Currently, most intensive care physicians use weaning parameters to predict the likelihood of successful extubation in ICU patients [1]. However, despite these pre-extubation evaluations, a significant proportion of patients still experienced extubation failure, defined as the need for reintubation within 72 hours after extubation.

In most clinical studies, approximately 10 to 20% of ICU patients will be reintubated within 72 hours after extubation [2–4]. Reintubation after extubation failure leads to complications such as cardiovascular events or ventilator-associated pneumonia and has been associated with an increase in mortality [3,5]. Extubation failure was also associated with increased morbidity, longer ICU length of stay, and increased medical cost [6–8]. Therefore, careful evaluation with different clinical tests, including spontaneous breathing trial (SBT) and cuff-leak test, was generally recommended before extubation to minimize the risk of extubation failure [9]. Several weaning parameters, such as the rapid shallow breathing index (RSBI), maximal inspiratory and expiratory pressure (MIP and MEP), have been shown to predict short-term outcomes after extubation [1].

Other risk factors for extubation failure have also been studied. In different clinical studies, old age, high Acute Physiology and Chronic Health Evaluation (APACHE II) score, longer duration of intubation and higher levels of positive end-expiratory pressure (PEEP) have been reported to be associated with an increased risk of extubation failures [10–12]. Underlying diseases, included chronic respiratory diseases, heart failure with reduced ejection fraction, and a lower Glasgow Coma Scale, were also associated with extubation failure [6,10,13–15]. Other clinical indicators including the $PaO_2$ to $FiO_2$ ratio (PF ratio), have also been studied but were only associated with extubation failure in some studies [16].

In this article, we aimed to examine possible clinical and demographic risk factors associated with extubation failure among ICU patients in a regional hospital in Taiwan.

## Material and methods

### Study patients and setting

This was a single-center retrospective cohort study at a regional hospital in Northern Taiwan equipped with 20 medical and 10 surgical ICU beds. Patients who were admitted to either the surgical or medical ICU between June 1st, 2016, and July 31st, 2018, and had received endotracheal tube insertion with mechanical ventilation for more than 72 hours were eligible for the study. Enrolled patients must be aged 18 years or older and undergo a planned extubation during their ICU stay. Patients who were considered inappropriate for extubation, had an

unplanned extubation (including self-extubation), or received tracheostomy after mechanical ventilation were excluded. For those who had multiple intubation and ICU admission during the study period, only the first planned extubation was enrolled in this study.

For all mechanically ventilated patients in our ICU, the indication for endotracheal intubation and mechanical ventilation were determined by the treating physician. As part of the routine practices, all patient would be evaluated for the appropriateness of endotracheal tube removal if all the following clinical criteria were met: (1) the vital signs were stabilized (defined as a temperature below 38 degree Celsius, systolic blood pressure above 90 mmHg, diastolic pressure above 60 mmHg and a heart rate between 60 to 100 beats per minute); (2) active problems were under control, determined by the attending physician; (3) using a low fractional inspired oxygen ($FiO_2$) below 0.4; (4) with acceptable oxygenation ($PaO_2 \geq 60mmHg$ or $SaO_2 \geq 90\%$) based on arterial blood gas analyses; and (5) using a low PEEP below 8cm $H_2O$. After the initial screening, a SBT and cuff-leak test were performed, and weaning parameters, include tidal volume ($V_T$), minute ventilation (MV), MIP, MEP, and RSBI were measured. The treating physician would then evaluate the results of these tests and determine if endotracheal tube removal could be attempted. This study included only patients who underwent planned extubation following the aforementioned practices. The ventilators used in our ICU during study period included Servo-S (Maquet, Germany), Puritan Bennett 840 (Puritan Bennett, Medtronic plc, the U.S.), Servo-I (Maquet, Germany), and Puritan Bennett 760 (Puritan Bennett, Medtronic plc, the U.S.).

## Study procedure

After enrollment, baseline characteristics, including demographics, indications for ICU admission, indications for mechanical ventilation, APACHE II score, laboratory results, the length of ICU stay, and the prescription of prophylactic corticosteroid to prevent post-extubation stridor before extubation, were retrospectively collected. Regarding body weight, the measurement was performed on the admission of the ICU, and the patients were classified into underweight (BMI below 18.5 kg/m$^2$), normal weight (BMI between 18.5 and 24.9 kg/m$^2$) and overweight (BMI above 25 kg/m$^2$) [17]. In this study, extubation failure was defined as those who required reintubation within the first 72 hours for any reason. The indication of reintubation and clinical outcomes were also recorded. The enrolled patients were then classified according to the outcome of extubation, and their baseline characteristics were compared to identify potential differences. The study was approved by the Ethical Review Committee of Min-Sheng General Hospital (registration number: MSIRB2018014) and the need for informed consent was waived due to the retrospective design.

## Statistical analysis

The demographics, clinical characteristics and weaning parameters were compared between those with and without extubation failure. Non-categorical variables were compared using Student's *t*-test or Mann–Whitney *U*-test, and categorical variables were compared using chi-square test or Fisher's exact test. To identify potential risk factors, univariate and multivariate logistic regression model were then performed. For multivariate analyses, a backward elimination process was used, in which all possible demographic and clinical factors were initially included in the model and stepwise removed starting with factors with the largest *p*-value. The process was repeated until all factors in the model had a *p* value of <0.2. All statistical analyses were performed using the Statistical Package for the Social Science version 25 (IBM Corporation, the USA). All *P*-values were two sided and a *P*-value of less than 0.05 indicated statistical significance.

## Results

During June 2016 and July 2018, 1,009 patients were intubated for the first time in the ICUs. After excluding patients who were intubated for less than 72 hours and other ineligible patients, 312 patients were evaluated for planned extubation during the study period (Fig 1). After the evaluation by the primary care ICU physician and respiratory therapist, 34 were considered unsuitable for extubation while 278 patients were ready for a planned extubation after mechanical ventilation for more than 72 hours. Finally, 10 patients did not undergo extubation due to various reasons after the evaluation, and 268 patients were included in our study for analysis. Overall, 19/268 (7.1%; 95% confidence interval [CI], 4.3–10.9%) patients failed their planned extubation, and these patients were all reintubated within 72 hours.

The most common causes of post-extubation failure were respiratory distress or stridor (12/19; 63.2%), followed by hypoxemia (4/19, 21.1%), hemodynamic instability (2/19, 10.1%), and hypercapnia (1/19, 5.3%). After reintubation, 42.1% (8/19) of patients were referred to a

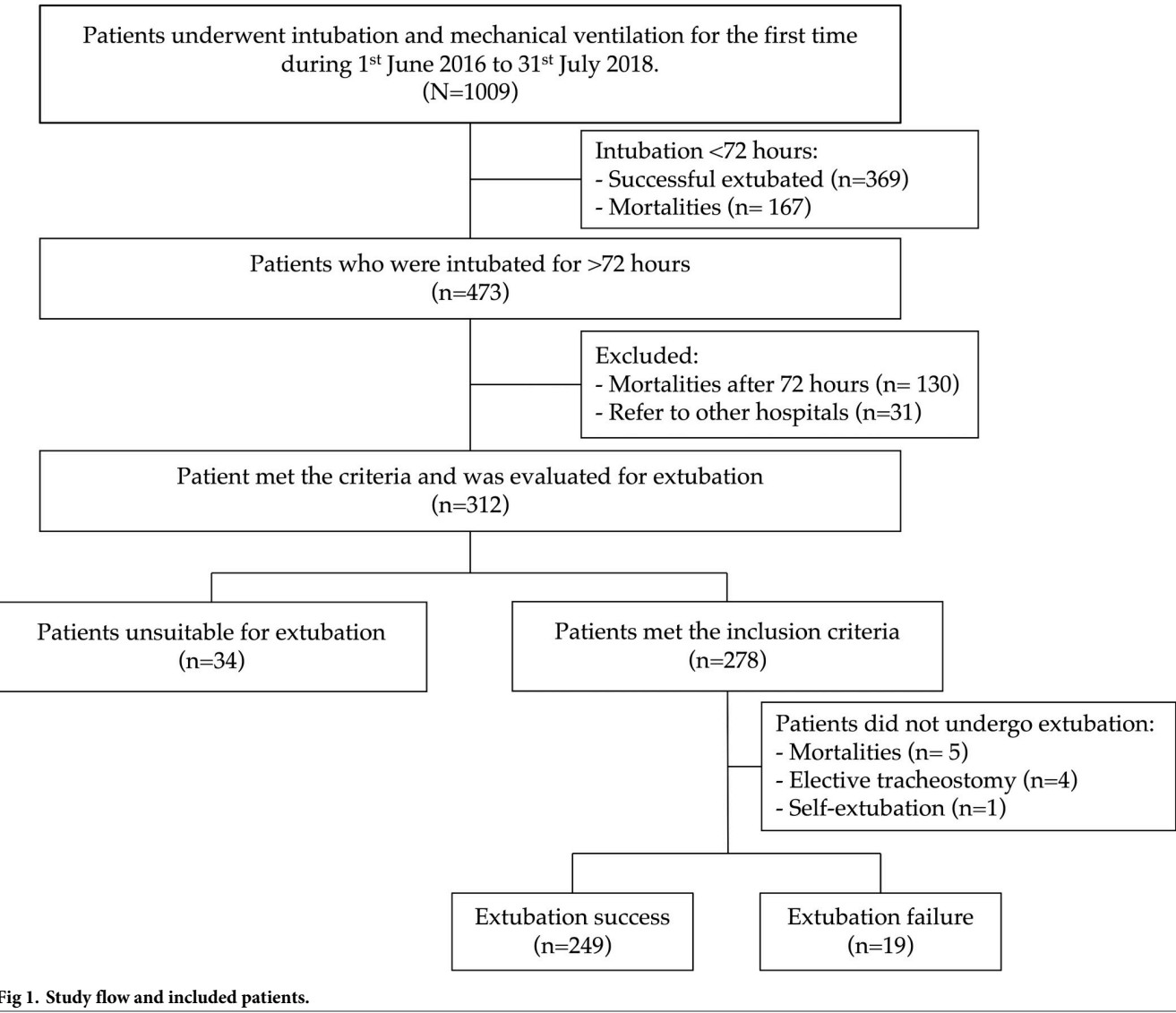

**Fig 1. Study flow and included patients.**

respiratory care center, 31.6% (6/19) of patients succeeded on their second attempt at planned extubation, 26.3% (5/19) died during subsequent ICU stays, and 15.7% (3/19) successfully weaned off ventilation after receiving elective surgical tracheostomy.

The baseline characteristics of included patients were listed in Table 1. Overall, 65.3% (175/268) of included patients were male with a median age of 67 years. In terms of the source of patients, 63.1% (169/268) of included patients were enrolled from medical ICU, and 36.9% (99/268) were from surgical ICU, which did not differ among the two groups. The duration of intubation before planned extubation was similar between those who succeeded and failed planned extubation (8 vs. 8 days, p = 0.904). When compared with those who had a successful extubation, those who failed on planned extubation were more likely to be underweight (42.1% vs. 16.9%, p = 0.01), had a significantly lower BMI (20.4 vs. 22.5 kg/m$^2$, p = 0.002), and longer ICU stay (22 vs. 12 days, p = 0.007).

**Table 1. Baseline characteristics of enrolled patients.**

| Baseline characteristics | All patients (N = 268) | Extubation success (N = 249) | Extubation failure (N = 19) | *P* value |
|---|---|---|---|---|
| Age, years, median (IQR) | 67 (55–80) | 67 (55–80) | 69 (61–73) | 0.66 |
| Male sex, n (%) | 175 (65.3) | 164 (65.9) | 11 (57.9) | 0.62 |
| BMI, kg/m$^2$, median (IQR) | 22.3 (19.3–26.0) | 22.5 (19.6–26.4) | 20.4 (16.1–22.3) | **0.002** |
| Underweight (BMI <18.5), n (%) | 50 (18.7) | 42 (16.9) | 8 (42.1) | **0.01** |
| Normal (BMI 18.5–24.9), n (%) | 135 (50.4) | 126 (50.6) | 9 (47.4) | |
| Overweight (BMI >25), n (%) | 83 (31.0) | 81 (32.5) | 2 (10.5) | |
| GCS <8, n (%) | 120 (44.8) | 111 (44.6) | 9 (47.4) | 0.81 |
| Smoker, n (%) | 91 (34.0) | 81 (32.9) | 9 (47.4) | 0.22 |
| Diabetes mellitus, n (%) | 96 (35.8) | 88 (35.3) | 8 (42.1) | 0.62 |
| Congestive heart failure, n (%) | 18 (6.7) | 17 (6.8) | 1 (5.2) | 1.00 |
| Chronic obstructive pulmonary diseases (COPD)†, n (%) | 34 (13.7) | 31 (12.4) | 3 (15.8) | 0.67 |
| End-stage renal disease (ESRD), n (%) | 24 (9.0) | 19 (7.7) | 5 (26.3) | **0.02** |
| Chronic kidney disease, n (%) | 30 (11.2) | 28 (11.2) | 2 (10.5) | 1.00 |
| Division | | | | 0.81 |
| Medical ICU, n (%) | 169 (63.1) | 158 (63.5) | 11 (57.9) | |
| Surgical ICU, n (%) | 99 (36.9) | 91 (36.5) | 8 (42.1) | |
| Indication of intubation, | | | | 0.99 |
| Shock, n (%) | 91 (34) | 84 (33.7) | 7 (36.8) | |
| Hypoxic respiratory failure, n (%) | 75 (35.4) | 89 (35.7) | 6 (31.6) | |
| Impaired consciousness, n (%) | 38 (14.2) | 35 (14.1) | 3 (15.8) | |
| Surgery, n (%) | 44 (16.4) | 41 (16.5) | 3 (15.8) | |
| APACHE II, score, median (IQR) | 20 (16–25) | 20 (16–25) | 20 (16–24) | 0.76 |
| ICU stay, days, median (IQR) | 12 (8–19) | 12 (8–18) | 22 (12–29) | 0.01 |
| Intubated within the first 24 hours of admission, n (%) | 233 (87.0) | 216 (86.7) | 17 (89.5) | 0.73 |
| Duration of intubation before planned extubation, days, median (IQR) | 8 (6–14) | 8 (6–14) | 8 (6–14) | 0.90 |
| Prophylactic use of corticosteroid before extubation‡, n (%) | 180 (67.2) | 166 (66.7) | 14 (73.7) | 0.62 |
| Serum Albumin level before extubation, g/dL, median (IQR) | 3.1 (2.6–3.6) | 3.1 (2.6–3.6) | 3.3 (2.4–3.8) | 0.66 |
| Serum potassium level before extubation, mEq/L, median (IQR) | 3.9 (3.5–4.4) | 3.9 (3.5–4.4) | 4.2 (3.6–4.5) | 0.28 |
| P/F ratio before extubation, median (IQR) | 336.5 (260.8–442.5) | 337 (260–446.5) | 313 (264–400) | 0.43 |

†Of the 34 patients with COPD, 9 patients had received systemic corticosteroid during the intubation period.

‡Defined as starting corticosteroid 24 hours before planned extubation to prevent post-extubation stridor.

**Table 2. Weaning parameters before planned extubation.**

| Baseline characteristics | All patients (N = 268) | Extubation success (N = 249) | Extubation failure (N = 19) | *P value* |
|---|---|---|---|---|
| Minute ventilation (MV), L/min, median (IQR) | 8.2 (6.7–10.2) | 8.3 (6.7–10.2) | 7.8 (6.0–10.2) | 0.44 |
| MV ≥10L/min, n (%) | 73 (27.2) | 67 (26.9) | 6 (31.2) | 0.66 |
| Tidal volume, mL, median (IQR) | 366 (309–463) | 366 (309–465) | 390 (309–400) | 0.51 |
| RSBI, median (IQR) | 61 (42–82.3) | 61 (42–83) | 64 (51–77.6) | 0.45 |
| Maximal expiratory pressure, $cmH_2O$, median (IQR) | 40 (27–50) | 40 (28–50) | 40 (26–50) | 0.68 |
| Maximal inspiratory pressure, $cmH_2O$, median (IQR) | 32 (26–40) | 32 (26–40) | 30 (18–35) | 0.06 |
| Cuff leak volume | | | | |
| <110 ml, n (%) | 53 (19.8) | 51 (20.5) | 2 (10.5) | 0.38 |

## Weaning parameters before planned extubation

All patients in our study passed the spontaneous breathing trial, underwent the cuff-leak test, and had their weaning parameters measured before planned extubation. The results of weaning parameters among patients with successful extubation and extubation failure were listed in Table 2. Among patients who succeeded or failed planned extubation in our cohort, there was no statistically significant difference in minute ventilation, tidal volume, rapid shallow breathing index, maximal expiratory airway pressure, or the results of cuff-leak test. Compared to those who succeeded in planned extubation, those who had failed in planned extubation in our cohort had a lower MIP (30 vs. 32 $cmH_2O$, respectively), with borderline statistical significance (p = 0.06).

## Factors associated with extubation failure

To identify potential factors associated with extubation failure, we performed univariate and multivariate logistic regression (Table 3). In the univariate analysis, having ESRD (Odds ratio [OR], 4.32; 95% CI, 1.41–13.3) and lower MIP (OR per one $cmH_2O$ decrease, 1.04; 95% CI 1.00–1.09) were linked with an increased risk of extubation failure, while being underweight

**Table 3. Univariate and multivariate logistic regression of risk factors associated with extubation failure.**

| Risk factors | Univariate analysis | Multivariate analysis |
|---|---|---|
| Age, per 1-year increase | 1.00 (0.97–1.03) | |
| Male sex, vs. female | 0.71 (0.28–1.84) | |
| BMI category, underweight vs. normal | 2.67 (0.97–7.35) | **3.80 (1.23–11.7)*** |
| overweight vs. normal | 0.35 (0.07–1.64) | 0.54 (0.11–2.79) |
| Smoking | 1.83 (0.72–4.69) | 2.61 (0.93–7.34) |
| Heart failure | 0.76 (0.10–6.03) | |
| End stage renal disease | **4.32 (1.41–13.3)*** | **6.62 (1.85–23.7)*** |
| Diabetes mellitus | 1.33 (0.52–3.43) | |
| Use of prophylactic corticosteroids before extubation | 1.40 (0.49–4.02) | |
| GCS <8 | 0.89 (0.35–2.28) | |
| Cuff-leak <110mL | 2.28 (0.49–9.78) | |
| Duration of intubation before extubation, per 1-day increase | 0.98 (0.91–1.06) | |
| RSBI >105 | 1.16 (0.25–5.32) | |
| Maximal inspiratory airway pressure, per 1-$cmH_2O$ decrease | **1.04 (1.00–1.09)*** | **1.05 (1.00–1.09)*** |

*Statistically significant.

had a borderline association (OR compared to normoweight patients, 2.67; 95% CI, 0.97–7.35). In the multivariate analysis, being underweight was significantly associated with extubation failure compared to those with a normal BMI (adjust OR [aOR], 3.80; 95% CI, 1.23–11.7). Other factors associated with extubation failure included lower MIP (aOR per one cmH$_2$O decrease, 1.05; 95% CI 1.00–1.09) and having ESRD (aOR, 6.62; 95%CI, 1.85–23.7).

To further understand the solo effect of being underweight, we performed an additional sensitivity analysis after excluding those with ESRD (n = 20) and those with a low MIP below 20 cmH$_2$O (n = 19). In the remaining 229 patients, the proportion of extubation failure was 3.8% (4/105), 12.8% (5/39), and 2.7% (2/74) for normoweight, underweight, and overweight patients, respectively. In the univariate logistic regression, being underweight remained borderline associated with extubation failure in the sensitivity analysis (OR compared to normoweight patients, 3.37; 95% CI, 0.86–13.2; p = 0.08).

## Discussion

This is a retrospective cohort study conducted at a single center involving 268 patients from surgical or medical ICU who underwent planned extubation after being intubated for more than 72 hours and passing a 30-minute spontaneous breathing trial. The overall proportion of patients who required reintubation after planned extubated was 7.1% (19/268; 95% CI, 4.3–10.9%) in our cohort, which was similar to previous observations in literature and reviews [18]. In the multivariate analysis, being underweight (aOR compared to those with normal BMI, 3.80; 95% CI, 1.23–11.7), having a lower MIP (aOR per one cmH$_2$O decrease, 1.05; 95% CI 1.00–1.09) and having ESRD (aOR, 6.62; 95%CI, 1.85–23.7) were independently associated with extubation failure.

The association between weight and clinical outcome during critical care had been widely discussed in the literature [19–21]. However, there is no direct evidence linking body weight with the risk of post-extubation respiratory failure among adult patients in the intensive care settings. The available evidence in the literature mostly focused on the relationship between weight, postoperative outcomes, and the overall outcome of critical care. In the literature, obesity and overweight have been repeatedly associated with deleterious outcome in critical care setting [19–22]. Besides, the difficulties in airway management among obese patients also increased the risk in the event of postoperative respiratory failure [23]. Other than obesity and its associated condition, extreme low body weight was also linked with poor outcome for critically ill patients. In a retrospective study including 1,488 mechanically ventilated patients, both obesity and underweight were significantly associated with increased hospital mortality [24]. In studies focusing on post-operative outcomes, cohorts from Iran and Germany had demonstrated that a lower BMI was associated with reintubation among patients who underwent coronary artery bypass grafting [25,26]. In a systemic review focusing on patients undergoing vascular surgery, underweight patients were associated with higher postoperative mortality and respiratory complication, including acute respiratory failure, pneumonia, failure to be weaned from ventilation postoperatively, and unplanned intubation [27]. However, to the best of our knowledge, the association between underweight and the failure of planned extubation after critical illness was yet described.

The pathophysiological change of underweight and respiratory function had been reported before. Lower BMI is associated with low protein and energy intake, malnutrition, cachexia, and inadequate nutritional reserve; and might result in the inability to compensate for the physiological stress of critical illness [28]. Underweight patients also have a lower lean muscle mass, which is subsequently associated with lower respiratory muscle mass, lower peak expiratory flow, and reduced hypoxic ventilatory response [29–33]. An animal study has shown how

nutritional deprivation reduced the contractility of diaphragm, the primary inspiratory muscle during respiration [34]. For ICU patients, studies have demonstrated the association between low BMI and reintubation, prolonged intubation time and ICU stay, which is likely related to respiratory muscle weakness and malnutrition [28,34]. Although these observations might provide explanation regarding the association between underweight and post-extubation respiratory failure among ICU patients, the pathophysiological link requires further investigation. For example, a number of medical conditions, including but not limited to malignancy, advanced lung diseases, uncontrolled HIV infection, alcoholism, vitamin deficiency, might be linked with the decrease in BMI as well as the outcome after extubation [28,35].

In our study, the MIP was measured before planned extubation, after the patient was considered suitable to extubate. Although it is not the best predictor, the association between MIP and weaning outcome had been repeatedly discussed before and the association is also reflected in our data [36,37]. In a study including 185 ICU patients, Jung et al. demonstrated the association between ICU-acquired weakness and diaphragm muscle dysfunction and its consequence of extubation failure [38]. As aforementioned, diaphragm muscle weakness has been linked with lower BMI and malnutrition in the literature [32,34], which could likely explain the association between underweight, low MIP and post-extubation respiratory failure in our study. Unfortunately, we were unable to perform physiological tests to demonstrate the function of diaphragm muscle among enrolled patients in our ICU. Furthermore, respiratory muscle weakness has been reported in a variety of settings, including cardiovascular disease [39], which might also contribute towards the respiratory failure after planned extubation.

Our study has several limitations and should be interpreted with caution. Although the routine practice to evaluate the appropriateness to extubate is well-protocolized in our hospital, this was still a retrospective study and the decision to extubate depended on the judgement of the physician in the ICU. Patients with better recovery and clinical condition were more likely to undergo extubation in clinical settings. Therefore, it was difficult to completely avoid confounding factors during the process despite the multivariate adjustment. Also, some clinical variables, such as serum phosphate level, total protein, cholesterol level, fluid status, etc., were not routinely recorded during the study period. Furthermore, sample size estimation was not performed during the design of study. A prospective study with adequate sample size is warranted to confirm our finding. Secondly, measuring height and weight in critical care setting might be challenging and biased by edema or fluid retention. Moreover, a simple measurement of BMI might not discriminate between lean muscle mass, fat mass, and fat distribution. Both the causes for underweight and low MIP is multifactorial, and thus the association needed to be validated in future studies. It is also important to notice that, in our study, most included patients underwent rigorous evaluation for their appropriateness of extubation and those who were unsuitable for extubation would be excluded (see Fig 1). Patients who were in poor clinically condition might not pass the evaluation and therefore excluded. Therefore, some of the traditional predictors, such as patients with high RSBI, low tidal volume, were not presented in our cohort. Our results demonstrated that underweight might be an additional risk factor among those underwent planned extubation, but the importance of other traditional risk factors should not be overlooked. Finally, since the weight distribution may differ between different countries or ethnicities, our results should be generalized with caution between different population groups.

## Conclusions

In conclusion, we demonstrated that for patients admitted to surgical or medical ICUs and underwent planned extubated after >72 hours of intubation, being underweight (aOR

compared to patients with normal weight, 3.80; 95% CI, 1.23–11.7), having a lower MIP (aOR per one $cmH_2O$ decrease, 1.05; 95% CI 1.00–1.09) and having ESRD (aOR, 6.62; 95%CI, 1.85–23.7) was associated with an elevated risk for post-extubation respiratory failure.

## Supporting information

**S1 Data.**
(XLS)

## Acknowledgments

Our thanks to the ICU staff in the Min-Sheng General Hospital (Taoyuan, Taiwan) for providing assistance throughout the study and data collection.

## Author Contributions

**Conceptualization:** Chung-Yeh Chuang, Han-Shui Hsu.

**Data curation:** Chung-Yeh Chuang, Han-Shui Hsu, Guan-Jhou Chen, Tzu-Yi Chuang, Ming-Han Tsai.

**Formal analysis:** Guan-Jhou Chen.

**Investigation:** Chung-Yeh Chuang.

**Methodology:** Chung-Yeh Chuang.

**Validation:** Chung-Yeh Chuang.

**Writing – original draft:** Chung-Yeh Chuang, Tzu-Yi Chuang, Ming-Han Tsai.

**Writing – review & editing:** Chung-Yeh Chuang, Han-Shui Hsu, Guan-Jhou Chen, Tzu-Yi Chuang, Ming-Han Tsai.

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
