## [Decision Letter · Decision Letter 0]

8 Dec 2022

PONE-D-22-30747Underweight Predicts Extubation Failure after Planned Extubation in Intensive Care UnitsPLOS ONE

Dear Dr. Chen,

Thank you for submitting your manuscript to PLOS ONE. After careful consideration, we feel that it has merit but does not fully meet PLOS ONE’s publication criteria as it currently stands. Therefore, we invite you to submit a revised version of the manuscript that addresses the points raised during the review process.

 Substantial revision of the manuscript in light of reviewer feedback is needed. Please respond to all reviewer comments.

We look forward to receiving your revised manuscript.

Kind regards,

Martin Kieninger

Academic Editor

PLOS ONE

Journal Requirements:

Reviewers' comments:

Reviewer's Responses to Questions

**Comments to the Author**

1. Is the manuscript technically sound, and do the data support the conclusions?

Reviewer #1: Yes

Reviewer #2: No

Reviewer #3: Yes

2. Has the statistical analysis been performed appropriately and rigorously? 

Reviewer #1: Yes

Reviewer #2: Yes

Reviewer #3: Yes

3. Have the authors made all data underlying the findings in their manuscript fully available?

Reviewer #1: Yes

Reviewer #2: No

Reviewer #3: Yes

4. Is the manuscript presented in an intelligible fashion and written in standard English?

Reviewer #1: Yes

Reviewer #2: No

Reviewer #3: Yes

5. Review Comments to the Author

Reviewer #1: This study evaluated the association between BMI and post-extubatgion failure in a single hospital. Overall, the study is well-designed and the manuscript is well-written. However, I haver several concerns.

Major concern

1. I have serious concern about the writing "lower body mass index (BMI) (adjust OR [aOR], 1.20 per 1-kg/m2 decrease; 95% CI, 1.05-1.37)". Based on this finding, overweight or obesity can be associated with a lower risk of extubation failure than normal weight. Treating body weigh as categorial variables (underweight vs normal weight) for further anlaysis may be more appropriate.

Minor concern

1. Please add the indication of MV in the table 1.

2. Please add some possible residual confounding factors - prior use of muscle relaxant, prophylaxis use of corticosteorid for prevention, phosphate level, and etc.

Reviewer #2: Dear authors, here you receive the review of the manuscript entitled ” Underweight Predicts Extubation Failure after Planned Extubation in Intensive Care Units”, Short Title: Underweight and extubation failure with ID: PONE-D-22-30747.

The authors present a retrospective cohort study in which they focused on the value of underweight of patients in the relation to extubation falure (N=268, initially 278).

Major remarks

Is this correct that there is not much to find in the literature with this negative focus?

As most studies may focus on the success of extubation and not the failure. Would is be possible to look at the success and find the absence of underweight as an important parameter?

The study was approved by an ethical review committee and the registration number can be found within the open access database (internet).

Description of the recruitment ( inclusion & exclusion) is provided.

Overall, 268 patients. Please when possible place this group, your cohort more in a representable way so readers may understand how these patients are well representable from the patients who were admitted to ICU during the study period?

The manuscript is easy to read and understand. However, the language and the many small textual mistakes , i.e., time, grammar, … may be improved by a critical reading from a native English speaker. I will pay attention to some, but there are too many!

Abstract:L3 “was less reported” Please clarify and be more precise by providing exact numbers , details and refs in articles , introduction and discussion?

what is meant by “aOR, 1.04 per 1-cmH2) decrease”, especially the “1-“? Please clarify or adapt?

Page19 LinL 8 please describe for how long the period or total time for hospital stay before extubation was for these patients? You can imagine theta for patients with an already longer stay in the hospital before operation or hospitalized on the wards, being bedridden or not able to move properly, muscle power was already diminished before the intubation or may have led to the indication of start of artificial ventilation.

P Secondly, measuring….fluid retention? This may be important as for instance total time of hospitalization, immobilization, time of sever sepsis, positive fluid balance, may have influenced a catabolic state that may interfere with the success of extubation. Please look in your data and regard these possible issues?

Also, the cause of post-extubation failure.. This remark is different from the expression on p8 “The indication of….. recorded”

P7 L15 what was in this study the meaning of the vital sign stabilized? Please describe exactly?

L16-17 (Pa O2≥60% or SaO2 ≥90%) PaO2 is a pressure and a percentage! Please adapt.

What is meant or quantified with (5) low PEEP? Is it ≤ 8 8 cm H2O?

I can imagine that patients were not included because of missing data , e.g. no cuff leak test , SBT, etc… Is that correct?

P8 LL13 corticosteroid? How long before extubation, So no patients with COPD and corticosteroids were included or where inhalational corticosteroids allowed. Please be more precise in your description around the term corticosteroids?

P8 here you describe that the indications for reintubation and clinical outcome were recorded. Where are they presented?

P8 and 9 please provide regarding the design of the study the power and argumentation, e.g. for number of patients needed to include?

P9 what was the overall % of reintubation of extubation failure in your department. So, considering the relative small cohort in relation to the total in your department. Is it a valuable and real representation? Please provide number and relate to your study results?

P10 L13 “… who succeed and failed…” What is meant by this quote? Is this for instance the cumulative number of days for the patients with failed extubation? So, these patients were extubated earlier and the time after reintubation and ventilation period was added tot the total time intubated?

P11/12 what were body temperatures and respiratory frequency, minute volume? AS these are known as relative strong predictors for failed extubation and need for reintubation <24 hours as you may imagine that lung function during illness at the moment is too small for increased metabolism (e.g. fever) and related to BMI.

P13 table 2, Please look differently and compare Minute ventilation< and > e.g.10L/min, and dynamic compliance? Here in this relative small group there is no significant difference. How ever when comparing patient with either or not minute volume >10 L/min this may be interesting?

Was there a difference between patients intubated for 72 hours vs patients for instance twice or 3 times this time-period? So, in other words what can be said by the time on ventilation in regard of the possibility of increased time on the ventilator?

P17 respiratory complication (22)? This is very vague for our readers. Please clarify what you mean?

L12 this also accounts for “reduced respiratory drive”? Did the patients have ICU-acquired weakness, (critical illness polyneuropathy) or what other?

L18 HIV infection, alcoholism, Vitamin deficiency . This remains very vague in what you mean in relation to your message and conclusion of your study? Please describe exactly when using these broad diagnoses. So, acute hiv-infection, therapy resistant infection, therapy non- compliance, etc, what specific vitamins and leading to what deficiencies and forms of organ failure. Why only HIV, or does may it account to aal kinds of viral or bacterial infections that are uncontrolled or treatment resistant and lead tot increase of metabolism?

P18 .. between low and.. what is meant with the word “low” ? please be exact?

The conclusion does not cover your results completely as you may see form the questions asked.

P23 Please be more precise in the presentation of your references? Take notice of the journal’s information to authors. e.g. capitals or not? The New England…

Minor remarks

Ad Abstract : page 3 Line 9 “The probability were” please change for “was”?

L 3 Textual remarks

Page 5 L 12 associated an change into associated with an …

P6 L4 to associated change into to be associated

L 6 failure change into failures

Brain injured change into brain injury And be more specific. All kinds of brain injury, a special kind of brain injury??

L8 conflicting results(14) please describe what is different or conflicting?

P7 L13 evaluation change into evaluated

P10 L5 panned change into planned

P16 L1 centered change into center

L4 spontaneous breathing. Here the word” trial” is missed

L16 challenge change into challenged

P17L6 and many other places in the text. post-operative change into postoperative

L11which subsequently change into which is subsequently

L12 Animal study change into An animal study has..

L16 condition change into conditions

P18 L1 study change into either “a study” as only 1 reference is provided (31) or into , studies..

P18 L3 Our study…. the association… Please change into We observed an association

L7 consider change into considered

L9 the associated change into the association

p 19 L 1 interpret change into Interpreted

L 3 This is a still retro… change into this still is….

L4 of physician change into of the

L5 is more likely change into are more likely

L12 to closely change into to be closely

Reviewer #3: 1. The authors indicated that low BMI is a risk for failed extubation Can you seek and show a cut-off for BMI?

2. Rehabilitation would be related to weaning of ventilation and extubation. Please add the information about the duration of rehabilitation.

3. In discussion section, the authors state that low BMI is associated with low protein and energy intake, malnutrition (Page 17). Please add the nutritional variables such as total protein, albumin, and cholesterol etc to Table 1.

6. PLOS authors have the option to publish the peer review history of their article (what does this mean?). If published, this will include your full peer review and any attached files.

Reviewer #1: No

Reviewer #2: **Yes: **p.bruins

Reviewer #3: No

---

## [Author Response · Author response to Decision Letter 0]

10 Feb 2023

Responses to Reviewer’s Comments

Reviewer #1:

Major concern:

I have serious concern about the writing "lower body mass index (BMI) (adjust OR [aOR], 1.20 per 1-kg/m2 decrease; 95% CI, 1.05-1.37)". Based on this finding, overweight or obesity can be associated with a lower risk of extubation failure than normal weight. Treating body weigh as categorial variables (underweight vs normal weight) for further analaysis may be more appropriate. 

Response:

Thank you for the comment. We agreed that the original presentation of data might be misleading and have ignored the risk associated with overweight. We treated the body weight as categorical variables and rebuilt the multivariate regression model. In the revised analysis, being underweight is significantly associated with extubation failure as compared with those with normal BMI (adjust OR [aOR], 3.80; 95% CI, 1.23-11.7) but not the overweight patients. Other factors associated with extubation failure included lower maximal inspiratory airway pressure (aOR, 1.05 per 1-cmH2O decrease; 95% CI 1.00-1.09) and having ESRD (aOR, 6.62; 95%CI, 1.85-23.7). The text in Results and Discussion were revised accordingly.

Minor concern:

1. Please add the indication of MV in the table 1.

Response:

Thank you for the query. Among the 268 enrolled patients, the most common indication of MV was shock (91/268, 34%), followed by hypoxic respiratory failure (75/268, 35.4%), surgery (44/268, 16.8%) and impaired consciousness (38/268, 14.2%). The information was updated in the Table 1.

2. Please add some possible residual confounding factors - prior use of muscle relaxant, prophylaxis use of corticosteroid for prevention, phosphate level, and etc.

Response:

Thank you for the suggestion. Unfortunately, due to the retrospective design of our study, some information, such as serum phosphate level, was not routinely collected during the study and were unavailable for analysis. We have amended the text in Discussion to mention this limitation.

 In terms of the use of corticosteroid, 180/268 (67.2%) of our patients received prophylactic corticosteroid prior to the planned extubation. However, the use of corticosteroid was not associated with extubation failure in both univariate and multivariate logistic regression model. The results were added in Table 3.

 

Reviewer #2:

Major remarks

1. Is this correct that there is not much to find in the literature with this negative focus? As most studies may focus on the success of extubation and not the failure. Would it be possible to look at the success and find the absence of underweight as an important parameter?

Response:

Thank you for the advice. However, to the best of our knowledge, there is no direct evidence linking body weight with post-extubation respiratory failure among adult patients in the intensive care settings. The available evidence in the literature mostly focused on the relationship between weight, post-operative outcomes, and the overall outcome of critical care. Furthermore, available large cohort studies investigating the outcome after extubation did not provide the weight of included subjects; therefore, it is difficult to establish any association between weight and post-extubation failure.

The study was approved by an ethical review committee and the registration number can be found within the open access database (internet). Description of the recruitment (inclusion & exclusion) is provided.

Overall, 268 patients. Please when possible place this group, your cohort more in a representable way so readers may understand how these patients are well representable from the patients who were admitted to ICU during the study period?

Response:

Thank you for the query. During June 2016 and July 2018, 1,009 patients were intubated for the first time and admitted in the ICUs. We then excluded patients who were intubated for less than 72 hours and who were otherwise ineligible (transferred to other facilities, etc.). A total of 312 patients were evaluated for planned extubation during study period (Fig. 1). After the evaluation by the primary care ICU physician and respiratory therapist, 34 were considered unsuitable for extubation while 278 patients were ready for a planned extubation after mechanical ventilation for more than 72 hours. Finally, 268 patients were included in our study for analysis.

 Among those who were suitable for a planned extubation after >72 of mechanical ventilation, only 10.9% (34/312) were considered unsuitable for extubation (mostly due to extreme old age or high demand of ventilation support). The text and Figure 1 were amended accordingly to include the study flow.

Abstract: L3 “was less reported” Please clarify and be more precise by providing exact numbers, details and refs in articles, introduction and discussion?

Response:

Thank you for the advice. As aforementioned, there is no direct evidence linking body weight with post-extubation respiratory failure among adult patients in the intensive care settings. The available evidence in the literature mostly focused on the relationship between weight, post-operative outcomes, and the overall outcome of critical care. 

Cohorts from Iran and Germany had demonstrated that a low BMI was associated with reintubation among patients who underwent coronary artery bypass grafting. Another systemic review focusing on patients undergoing vascular surgery also demonstrated that underweight were associated with higher postoperative mortality and respiratory complication, including acute respiratory failure, pneumonia, failure to be weaned from ventilation postoperatively, and unplanned intubation. We had amended the Discussion to include the details.

What is meant by “aOR, 1.04 per 1-cmH2) decrease”, especially the “1-“? Please clarify or adapt?

Response:

We have amended the text to “adjust odds ratio per one cmH2O decrease”, thank you.

Page19 Line 8 please describe for how long the period or total time for hospital stay before extubation was for these patients? You can imagine theta for patients with an already longer stay in the hospital before operation or hospitalized on the wards, being bedridden or not able to move properly, muscle power was already diminished before the intubation or may have led to the indication of start of artificial ventilation.

Response:

 In our cohort, 87% (233/268) patients where intubated upon admission or within the first 24 hours of admission. There was no difference between those who succeed or failed planned extubation. It is likely that patients who were bedridden or chronically ill might be considered unsuitable for extubation and excluded from the cohort. We had included the information in Table 1 and the Discussion.

Secondly, measuring…. fluid retention? This may be important as for instance total time of hospitalization, immobilization, time of sever sepsis, positive fluid balance, may have influenced a catabolic state that may interfere with the success of extubation. Please look in your data and regard these possible issues?

Response:

Thank you for the query. Unfortunately, due to the retrospective design of our study, some information, such as the change of weight (which is a good indicator for fluid status), was not routinely collected during the study and were unavailable for analysis. However, in the day-to-day practice in our ICU, patients with extreme fluid retention were unlikely to be evaluated for extubation under the conditions associated with fluid retention was corrected. We have amended the text in Discussion to mention this limitation.

Also, the cause of post-extubation failure. This remark is different from the expression on p8 “The indication of…. recorded”

Response:

In our study, the most common causes of post-extubation failure were respiratory distress or stridor (12/19; 63.2%), followed by hypoxemia (4/19, 21.1%), respiratory distress (2/10, 10.1%), hemodynamic instability (2/19, 10.1%), and hypercapnia (1/19, 5.3%). The information was added in the Result section.

P7 L15 what was in this study the meaning of the vital sign stabilized? Please describe exactly?

Response:

Thank you for the query. In the study, stable vitals were defined as temperature below 38 degree Celsius, systolic blood pressure above 90 mmHg, diastolic pressure above 60 mmHg and a heart rate between 60 to 100 beats per minute. The details were included in the Method section.

L16-17 (Pa O2≥60% or SaO2 ≥90%) PaO2 is a pressure and a percentage! Please adapt.

Response: The text was amended accordingly. Thank you!

What is meant or quantified with (5) low PEEP? Is it ≤ 8 cm H2O?

I can imagine that patients were not included because of missing data , e.g. no cuff leak test , SBT, etc. Is that correct?

Response:

Thank you for the query. In the study, we defined low PEEP as those who were using a PEEP level below 8-cmH2O. Although our study was retrospective, but the record of PEEP level and settings of mechanical ventilation is part of standard medical records in Taiwan; so there is no patients who were excluded due to missing data during the study period.

P8 L13 corticosteroid? How long before extubation, so no patients with COPD and corticosteroids were included or where inhalational corticosteroids allowed. Please be more precise in your description around the term corticosteroids?

Response:

Among the enrolled patients, 34 had COPD before intubation and 26.5% (9/34) of these patients had received systemic corticosteroid during intubation. On the other hand, use of prophylactics corticosteroid was defined as those who were started on corticosteroid within the 24 hours before planned extubation, aiming to prevent post-extubation stridor. We had amended the text in Method section and Table 1 to clarify the definition.

P8 here you describe that the indications for reintubation and clinical outcome were recorded. Where are they presented?

Response:

Thank you for the query. In our study, the most common causes of post-extubation failure were respiratory distress or stridor (12/19; 63.2%), followed by hypoxemia (4/19, 21.1%), hemodynamic instability (2/19, 10.1%), and hypercapnia (1/19, 5.3%). After the reintubation, 42.1% (8/19) of patients were referred to a respiratory care center, 31.5% (6/19) patients succeed on their second attempt of planned extubation, 26.3% (5/19) died during subsequent ICU stays, and 15.7% (3/19) successfully weaned off ventilation after receiving elective surgical tracheostomy. The information was updated in the Result section. 

P8 and 9 please provide regarding the design of the study the power and argumentation, e.g. for number of patients needed to include?

Response:

Thank you for the query. Limited by the retrospective design, we did not perform sample size estimation before initiating the study, but we attempted to include all available patients during the study period for analysis. The text of discussion is revised to highlight this particular limitation of study design.

P9 what was the overall % of reintubation of extubation failure in your department. So, considering the relatively small cohort in relation to the total in your department. Is it a valuable and real representation? Please provide number and relate to your study results.

Response:

Thank you for the query. During June 2016 and July 2018, 1,009 patients were intubated for the first time and admitted in the ICUs. We then excluded patients who were intubated for less than 72 hours and who were otherwise ineligible (transferred to our facilities, etc.). A total of 312 patients were evaluated for planned extubation during study period (Fig. 1). After the evaluation by the primary care ICU physician and respiratory therapist, 34 were considered unsuitable for extubation while 278 patients were ready for a planned extubation after mechanical ventilation for more than 72 hours. Finally, 268 patients were included in our study for analysis.

 Since most patients who underwent planned extubation during the study period were included, we believed our cohort was representative of the patients who sought medical attention in our hospital. However, our hospital is a local hospital operating in Northern Taiwan and the results might not be generalized to other geographical area or populations. The text and Figure 1 were amended accordingly to include the study flow.

P10 L13 “… who succeed and failed…” What is meant by this quote? Is this for instance the cumulative number of days for the patients with failed extubation? So, these patients were extubated earlier and the time after reintubation and ventilation period was added tot the total time intubated?

Response:

Here we were discussing the duration of intubation before planned extubation, time after reintubation was not included. The text was amended to avoid confusion.

P11/12 what were body temperatures and respiratory frequency, minute volume? AS these are known as relative strong predictors for failed extubation and need for reintubation <24 hours as you may imagine that lung function during illness at the moment is too small for increased metabolism (e.g. fever) and related to BMI.

Response:

Thank you for the comment. Indeed, fever and other physiological parameters were known predictors to extubation failure. In our practice, patients with active fever would be postponed for extubation until the cause of fever was confirmed and temperature controlled. This is an important criterion when we evaluate the appropriateness of extubation. The text was amended to include this information (in Method section). We also compared the minute ventilation and tidal volume of those who succeed and failed planned extubation, which was similar in both groups.

 It is also interesting to notice that, in our study, these “well-known” risk factors appeared to have no impact on the outcome of extubation. The most likely explanation is that our patients already underwent rigorous evaluation for the appropriateness (including spontaneous breathing trial, measurement of weaning parameters, etc.) before proceeding to extubation. Therefore, those in extreme condition and unsuitable for extubation were excluded beforehand (see Figure 1). However, our results should not diminish the importance of other risk factors. We have included this explanation in the Discussion to avoid misinterpretation of our results.

P13 table 2, Please look differently and compare Minute ventilation< and > e.g.10L/min, and dynamic compliance? Here in this relatively small group there is no significant difference. However, when comparing patient with either or not minute volume >10 L/min this may be interesting? 

Response:

In our cohort, there was 27.2% (73/268) patients with MV ≥ 10L/min, which was not differed among those who succeed or failed extubation (26.9% [67/249] vs. 31.2% [6/19], p=0.66). The results were added in Table 2. We also attempted to included MV in the multivariate logistic regression model as a categorical variable (≥10L/min vs. <10L/min), but it was not associated with the risk of extubation failure (data not shown).

Was there a difference between patients intubated for 72 hours vs patients for instance twice or 3 times this time-period? So, in other words what can be said by the time on ventilation in regard of the possibility of increased time on the ventilator?

Response:

Thank you for the query. In the univariate and multivariate logistic regression, in which the duration of intubation was treated as a continuous variable, there was no significant difference associated with the duration of intubation. We also built an additional model, treating the duration of intubation as categorical variable (3-6 days vs. 6-9 days vs. >9 days), and there was no statistically significant association noted (data not shown).

P17 respiratory complication (22)? This is very vague for our readers. Please clarify what you mean?

Response: The text was amended accordingly. In the reference cited, it described respiratory complications including acute respiratory failure, pneumonia, failure to be weaned from ventilation at 48 hours postoperatively, and unplanned intubation.

L12 this also accounts for “reduced respiratory drive”? Did the patients have ICU-acquired weakness, (critical illness polyneuropathy) or what other?

Response: We have amended the text into “……associated with lower respiratory muscle mass, lower peak expiratory flow, and reduced hypoxic ventilatory response.” Thank you.

L18 HIV infection, alcoholism, Vitamin deficiency. This remains very vague in what you mean in relation to your message and conclusion of your study? Please describe exactly when using these broad diagnoses. So, acute hiv-infection, therapy resistant infection, therapy non- compliance, etc, what specific vitamins and leading to what deficiencies and forms of organ failure. Why only HIV, or does may it account to aal kinds of viral or bacterial infections that are uncontrolled or treatment resistant and lead tot increase of metabolism?

Response:

Thank you for the suggestion. In the paragraph we meant to explain that, underweight might be associated with a myriad of clinical condition, including but not limited to limited to malignancy, advanced lung diseases, uncontrolled HIV infection, alcoholism, vitamin deficiency. All these clinical conditions might be associated with extubation failure by themselves. Therefore, the explanation and interpretation of our data should be done cautiously. We have reconstructed and rephrased the paragraph for better understanding.

P18 .. between low and.. what is meant with the word “low” ? please be exact?

The conclusion does not cover your results completely as you may see form the questions asked.

Response:

Thank you for the suggestion. We have changed the multivariate analysis and treated the BMI as categorical variables. In the new analysis, being underweight (using the WHO definition), is associated with an 3.80 increase of risk of extubation failure when compared to those with normal weight (95% CI, 1.23-11.7).

P23 Please be more precise in the presentation of your references? Take notice of the journal’s information to authors. e.g. capitals or not? The New England…

Response:

The listing of reference is revised to match the style of journal.

Minor remarks 

Abstract: page 3 Line 9 “The probability were” please change for “was”?

L3 Textual remarks

P5 L12 associated an change into associated with an …

P6 L4 to associated change into to be associated

L6 failure change into failures

Brain injured change into brain injury and be more specific. All kinds of brain injury, a special kind of brain injury??

L8 conflicting results (14) please describe what is different or conflicting?

P7 L13 evaluation change into evaluated

P10 L5 panned change into planned

P16 L1 centered change into center

L4 spontaneous breathing. Here the word” trial” is missed

L16 challenge change into challenged

P17 L6 and many other places in the text. post-operative change into postoperative

L11 which subsequently change into which is subsequently

L12 Animal study change into An animal study has..

L16 condition change into conditions

P18 L1 study change into either “a study” as only 1 reference is provided (31) or into , studies..

P18 L3 Our study…. the association… Please change into We observed an association

L7 consider change into considered

L9 the associated change into the association

P19 L1 interpret change into Interpreted

L 3 This is a still retro… change into this still is….

L4 of physician change into of the

L5 is more likely change into are more likely

L12 to closely change into to be closely

Response:

Thank you for the correction. All these remarks were amended accordingly.

 

Reviewer #3: 

The authors indicated that low BMI is a risk for failed extubation Can you seek and show a cut-off for BMI?

Response:

Thank you for the query. In our study, a BMI below 18.5 kg/m2 was defined as underweight, while BMI between 18.5 to 24.9 was defined as normal and BMI above 25 was defined as overweight. This definition was suggested by the WHO. We have specified the information in the Method section for better understanding.

Rehabilitation would be related to weaning of ventilation and extubation. Please add the information about the duration of rehabilitation.

Response:

Thank you for the query. Our study was performed in an acute setting, which means most patients were evaluated for extubation after the condition resulted in respiratory failure was controlled. Therefore, most of our patients were not engaged in rehabilitation training during their ICU stay.

In discussion section, the authors state that low BMI is associated with low protein and energy intake, malnutrition (Page 17). Please add the nutritional variables such as total protein, albumin, and cholesterol etc. to Table 1.

Response:

Thank you for the query. Unfortunately, due to the retrospective design of our study, some information, such as total protein, cholesterol, was not routinely collected during the study and were unavailable for analysis. In terms of the serum albumin level, there was no significant different between those who succeed or failed planned extubation (median, 3.1 vs. 3.3 g/dL, respectively; p=0.66). We have amended the text in Discussion to mention this limitation

---

## [Decision Letter · Decision Letter 1]

14 Mar 2023

PONE-D-22-30747R1Underweight Predicts Extubation Failure after Planned Extubation in Intensive Care UnitsPLOS ONE

Dear Dr. Chen,

Thank you for submitting your manuscript to PLOS ONE. After careful consideration, we feel that it has merit but does not fully meet PLOS ONE’s publication criteria as it currently stands. Therefore, we invite you to submit a revised version of the manuscript that addresses the points raised during the review process.

We look forward to receiving your revised manuscript.

Kind regards,

Martin Kieninger

Academic Editor

PLOS ONE

Journal Requirements:

Reviewers' comments:

Reviewer's Responses to Questions

**Comments to the Author**

1. If the authors have adequately addressed your comments raised in a previous round of review and you feel that this manuscript is now acceptable for publication, you may indicate that here to bypass the “Comments to the Author” section, enter your conflict of interest statement in the “Confidential to Editor” section, and submit your "Accept" recommendation.

Reviewer #1: All comments have been addressed

Reviewer #2: All comments have been addressed

Reviewer #3: All comments have been addressed

2. Is the manuscript technically sound, and do the data support the conclusions?

Reviewer #1: Yes

Reviewer #2: Yes

Reviewer #3: Yes

3. Has the statistical analysis been performed appropriately and rigorously? 

Reviewer #1: Yes

Reviewer #2: Yes

Reviewer #3: Yes

4. Have the authors made all data underlying the findings in their manuscript fully available?

Reviewer #1: Yes

Reviewer #2: Yes

Reviewer #3: Yes

5. Is the manuscript presented in an intelligible fashion and written in standard English?

Reviewer #1: Yes

Reviewer #2: Yes

Reviewer #3: Yes

6. Review Comments to the Author

Reviewer #1: The authors response well, and the manuscript was revised according. Therefore, I have no more comments.

Reviewer #2: Dear authors,

Here you receive my 2 nd review of the manuscript entitled “Underweight Predicts Extubation Failure after Planned Extubation in Intensive Care Units “with short title: “Underweight and extubation failure “ and ID: PONE-D-22-30747R1.

The authors took our previous comments very seriously in trying to improve understanding of their study results. However, I have some questions. When you answer one of our questions regarding lower body mass, vs overweight you found that besides underweight other factors more related tot ventilation were associated with extubation failure, e.g. decreased maximal inspiratory airway pressure (aOR, 1.05 per 1-cmH2O decrease; 95% CI 1.00-1.09) and having ESRD. Did you find any association between relative underweight and decrease of maximal inspiration pressure and/or ESRD. Ans also in other words, did patients with underweight within the population and no ESRD and normal maximal inspiration pressure also show an increased risk of extubation failure?

In my opinion the results are presented in a better understandable way and also their conclusions are improved, by making possible other confounders more visible and proving more weight to the the “underweight” aspects of their study, such as for instance ” We have changed the multivariate analysis and treated the BMI as categorical variables. In the new analysis, being underweight (using the WHO definition), is associated with an 3.80 increase of risk of extubation failure when compared to those with normal weight (95% CI, 1.23-11.7). “

For instance, also table 3 now provides better valuable information supporting the conclusion.

Minor remarks

P6 “…and a lower Glasgow come scale …“ change come into coma

P11 “…succeed on their second attempt” change succeed into past tense succeed

P16 “…while being underweight was had borderline association (OR compared to normoweight patients, 2.67; 95% CI, 0.97-7.35)” remove here the word “was”

Ad Table 3 please provide in a legend or other remark what is meant with the asterisk * within the table?

P19 post-operative may be written as postoperative

P23 “…might differ across among different…”change sentence into “Finally, since the weight distribution may differ between different countries or ethnicities, our results should be generalized with caution between different population groups.”

For instance Ref 25=ref35 please check total list for possible doubles and the right presentation?

Moreover, this reference is not numbered! “Nemer SN, Barbas CS, Caldeira JB, Guimarães B, Azeredo LM, Gago R, et al. Evaluation of maximal inspiratory pressure, tracheal airway occlusion pressure, and its ratio in the weaning outcome. J Crit Care. 2009;24(3):441-6.”

Reviewer #3: (No Response)

7. PLOS authors have the option to publish the peer review history of their article (what does this mean?). If published, this will include your full peer review and any attached files.

Reviewer #1: No

Reviewer #2: **Yes: **P.Bruins

Reviewer #3: No

---

## [Author Response · Author response to Decision Letter 1]

17 Mar 2023

Responses to Reviewer’s Comments

Reviewer #2:

Here you receive my 2nd review of the manuscript entitled “Underweight Predicts Extubation Failure after Planned Extubation in Intensive Care Units “with short title: “Underweight and extubation failure“ and ID: PONE-D-22-30747R1.

The authors took our previous comments very seriously in trying to improve understanding of their study results. However, I have some questions. When you answer one of our questions regarding lower body mass, vs overweight you found that besides underweight other factors more related total ventilation were associated with extubation failure, e.g. decreased maximal inspiratory airway pressure (aOR, 1.05 per 1-cmH2O decrease; 95% CI 1.00-1.09) and having ESRD. Did you find any association between relative underweight and decrease of maximal inspiration pressure and/or ESRD. Ans also in other words, did patients with underweight within the population and no ESRD and normal maximal inspiration pressure also show an increased risk of extubation failure?

Response:

Thank you for the query. To exam the association and potential multicollinearity between underweight, having ESRD and having a low MIP level, we calculated the variance inflation factor (VIF) of the final multivariate logistic model and no potential multicollinearity was detected.

Variables Variance Inflation Factor

Underweight vs. normoweight 1.29

Overweight vs. normoweight 1.66

Maximal inspiratory pressure 2.51

Smoking 1.52

Having ESRD 1.09

Furthermore, as an additional sensitivity analysis, we excluded those with ESRD (n=20) and those with a MIP<20 cmH2O (n=19). In the remaining 229 patients, 105 were normoweight and the proportion of extubation failure was 3.8% (4/105) while the proportion of extubation failure were 12.8% (5/39) and 2.7% (2/74) for underweight and overweight patients, respectively. In the univariate logistic regression, being underweight remained borderline associated with extubation failure (OR compared to normoweight patients, 3.37; 95% CI, 0.86-13.2; p=0.08). This additional analysis were added in the result section.

-----

In my opinion the results are presented in a better understandable way and also their conclusions are improved, by making possible other confounders more visible and proving more weight to the the “underweight” aspects of their study, such as for instance ” We have changed the multivariate analysis and treated the BMI as categorical variables. In the new analysis, being underweight (using the WHO definition), is associated with an 3.80 increase of risk of extubation failure when compared to those with normal weight (95% CI, 1.23-11.7). “

For instance, also table 3 now provides better valuable information supporting the conclusion.

Response:

Thank you for the comments and helping us improving the manuscript.

-----

Minor remarks

P6 “…and a lower Glasgow come scale …“ change come into coma

P11 “…succeed on their second attempt” change succeed into past tense succeed

P16 “…while being underweight was had borderline association (OR compared to normoweight patients, 2.67; 95% CI, 0.97-7.35)” remove here the word “was”

Ad Table 3 please provide in a legend or other remark what is meant with the asterisk * within the table?

P19 post-operative may be written as postoperative

P23 “…might differ across among different…”change sentence into “Finally, since the weight distribution may differ between different countries or ethnicities, our results should be generalized with caution between different population groups.”

Response:

Thank you for the correction. The text was amended accordingly.

-------

In the reference 

For instance Ref 25 = Ref 35, please check total list for possible doubles and the right presentation?

Moreover, this reference is not numbered! “Nemer SN, Barbas CS, Caldeira JB, Guimarães B, Azeredo LM, Gago R, et al. Evaluation of maximal inspiratory pressure, tracheal airway occlusion pressure, and its ratio in the weaning outcome. J Crit Care. 2009;24(3):441-6.”

Response:

Thank you for the correction. We have rechecked the order and listing of references.

---

## [Decision Letter · Decision Letter 2]

4 Apr 2023

Underweight Predicts Extubation Failure after Planned Extubation in Intensive Care Units

PONE-D-22-30747R2

Dear Dr. Chen,

We’re pleased to inform you that your manuscript has been judged scientifically suitable for publication and will be formally accepted for publication once it meets all outstanding technical requirements.

Kind regards,

Martin Kieninger

Academic Editor

PLOS ONE

Additional Editor Comments (optional):

Reviewers' comments:

Reviewer's Responses to Questions

**Comments to the Author**

1. If the authors have adequately addressed your comments raised in a previous round of review and you feel that this manuscript is now acceptable for publication, you may indicate that here to bypass the “Comments to the Author” section, enter your conflict of interest statement in the “Confidential to Editor” section, and submit your "Accept" recommendation.

Reviewer #2: All comments have been addressed

2. Is the manuscript technically sound, and do the data support the conclusions?

Reviewer #2: Yes

3. Has the statistical analysis been performed appropriately and rigorously? 

Reviewer #2: Yes

4. Have the authors made all data underlying the findings in their manuscript fully available?

Reviewer #2: Yes

5. Is the manuscript presented in an intelligible fashion and written in standard English?

Reviewer #2: Yes

6. Review Comments to the Author

Reviewer #2: Dear authors, here you receive my 2nd review regarding the manuscript entitled ”Underweight Predicts Extubation Failure after Planned Extubation in Intensive Care Units” with ID: PONE-D-22-30747R2.

In response to our criticism, the authors have thoroughly revised the manuscript and modified it where necessary. This contributes to improved readability and understanding of the paper. I have no further comments.

7. PLOS authors have the option to publish the peer review history of their article (what does this mean?). If published, this will include your full peer review and any attached files.

Reviewer #2: **Yes: **p.bruins

---

## [Editor Report · Acceptance letter]

6 Apr 2023

PONE-D-22-30747R2 

Underweight Predicts Extubation Failure after Planned Extubation in Intensive Care Units 

Dear Dr. Chen:

I'm pleased to inform you that your manuscript has been deemed suitable for publication in PLOS ONE. Congratulations! Your manuscript is now with our production department. 

Kind regards, 

on behalf of

Dr. Martin Kieninger 

Academic Editor

PLOS ONE